# Enhanced Coprime Array Structure and DOA Estimation Algorithm for Coherent Sources

**DOI:** 10.3390/s24010260

**Published:** 2024-01-02

**Authors:** Xiaolei Han, Xiaofei Zhang

**Affiliations:** 1Institute of IoT Engineering, Shanghai Business School, Shanghai 201400, China; 2Electronic Information College, Nanjing University of Aeronautics and Astronautics, Nanjing 210016, China; zhangxiaofei@nuaa.edu.cn

**Keywords:** enhanced coprime array, direction of arrival (DOA) estimation, coherent signals, array processing, spatial smoothing

## Abstract

This paper presents a new enhanced coprime array for direction of arrival (DOA) estimation. Coprime arrays are capable of estimating the DOA using coprime properties and outperforming uniform linear arrays. However, the associated algorithms are not directly applicable for estimating the DOA of coherent sources. To overcome this limitation, we propose an enhanced coprime array in this paper. By increasing the number of array sensors in the coprime array, it is feasible to enlarge the aperture of the array and these additional array sensors can be utilized to achieve spatial smoothing, thus enabling estimation of the DOA for coherent sources. Additionally, applying the spatial smoothing technique to the signal subspace, instead of the conventional spatial smoothing method, can further improve the ability to reduce noise interference and enhance the overall estimation result. Finally, DOA estimation is accomplished using the MUSIC algorithm. The simulation results demonstrate improved performance compared to traditional algorithms, confirming its feasibility.

## 1. Introduction

Extensive research has been conducted on the topic of direction of arrival (DOA) estimation in many array processing areas [1,2,3,4]. Antenna arrays are frequently used in these sectors to receive incoming signals. For instance, DOA estimation that impinges on an array with unknown mutual coupling is important in the area of navigation [5,6]. Scholars have proposed various DOA estimation algorithms using uniform linear arrays (ULAs), such as the ROOT-MUSIC technique [7], propagator method (PM) [8], CAPON [9], multiple signal classification (MUSIC) [10], and estimation of signal parameters via rotational invariance techniques (ESPRIT) [11]. Nevertheless, it is important to recognize that these techniques are predicated on the idea that the signals are independent of one another. Multipath propagation actually contributes to the coherent reception of signals. Because of this coherence, there is a rank deficiency in the covariance matrix of the received signals, which prevents these approaches from accurately estimating the DOA.

To solve the rank deficiency in the covariance matrix of received signals, a lot of approaches have been designed. One such approach is the generalized MUSIC algorithm, introduced in [12]. Another method involves the subspace adaptation technique, as presented in [13]. However, it is important to note that these methods entail multidimensional searching, making them computationally complex and impractical for real-world applications. The spatial smoothing technique [14] has undergone improvements in subsequent research (commonly referred to as MSSP), as documented in [15,16,17]. By combining the covariance matrices of each overlapping subarray, the rank of the covariance matrix is recovered by treating the entire array as a set of overlapping subarrays. Spatial smoothing-based approaches have been widely used and have undergone further enhancements [18,19].

Several alternative approaches have been developed for reconstructing the covariance matrix of coherent signals apart from spatial smoothing. Notably, novel matrix construction methods proposed in [20,21] do not require the signals to be independent. To recover the rank, the Toeplitz matrix construction is presented in [20], whereby the array aperture is sacrificed. On the other hand, ref. [21] presents a non-Toeplitz matrix approach that can manage scenarios involving coherent and incoherent signals. Improved spatial smoothing (ISS) is another noteworthy method that has attracted a lot of interest [22,23]. To improve the effectiveness of DOA estimation, these techniques make use of both the cross-covariance matrix between subarrays and the covariance matrix of each subarray. Additionally, ref. [24] presents an improved spatial smoothing method that is robust to noise and further utilizes information concerning signal subspace.

However, it is crucial to remember that current methods are made for uniform linear arrays (ULAs) and might not work well with sparse arrays, including coprime arrays (CA) [25,26,27] and nested arrays (NA) [28]. Sparse arrays have bigger aperture arrays than ULAs, and the inter-sensor spacing is not limited to half a wavelength. These sparse arrays therefore offer improved performance and create new opportunities for DOA estimation.

The virtual array faces limitations when it comes to processing DOA estimation for coherent sources with sparse arrays, thereby constraining the potential advantages of increased DOA design flexibility. Nevertheless, there are compressed sensing algorithms available for the direct estimation of coherent signals. A sparsity-aware algorithm is presented in [29] for reconstructing the covariance matrix using cyclic rank minimization. The nuclear norm is employed in the construction of the covariance matrix [30], leveraging its positive semi-definite (PSD) structure. A gridless algorithm, utilizing nuclear norm minimization (NNM) for interpolation, is introduced in [31]. Ref. [32] proposes an array interpolation algorithm that utilizes atomic norm minimization (ANM). Sparse Bayesian learning (SBL) [33] provides an efficient method for estimating a sparse signal, which determines the set of DOAs that have non-zero source power from a larger set of potential DOAs. While these algorithms offer high DOA estimation accuracy, it is important to note that they demand a substantial number of sensors and entail relatively high computational complexity.

Although the above compressed sensing algorithms can be directly used for the DOA estimation of coherent sources in sparse arrays, they cannot take full advantage of some characteristics of sparse arrays, such as the ambiguity resolution property of coprime linear arrays, and the performance of DOA estimation needs to be improved. In this paper, an enhanced coprime array is proposed by expanding sensors of the coprime array, combining the enhanced spatial smoothing algorithm and the characteristics of the mutual prime array to eliminate angle ambiguity. The enhanced spatial smoothing is performed on each subarray, and the DOA is estimated using the MUSIC algorithm. Finally, the angle ambiguity is eliminated via the mutual prime characteristics. The computation complexity is lower than compressed sensing algorithms, and the estimation accuracy of DOA is higher than compressed sensing algorithms, which is more suitable for practical application. Specifically, the following succinctly describes our primary contributions:We designed an enhanced coprime array with a larger array aperture that can be used to estimate the DOA of coherent sources;We utilized an enhanced spatial smoothing technique to effectively smooth the signal subspace rather than directly smoothing the received signal. This technique enhances the capability to resist noise interference and greatly improves the performance of DOA estimation.We conducted the performance comparison between the proposed algorithms and classical compressive sensing algorithms for DOA estimation of coherent sources, which demonstrated the superior performance of the proposed algorithms, along with significantly lower computational complexity compared to compressive sensing algorithms.

The remainder of this paper is provided below: Section 2 presents how to design the enhanced coprime array. Section 3 analyzes how the spatial smoothing algorithm and enhanced spatial smoothing algorithm perform the decorrelation step, demonstrating how coprime arrays are utilized to eliminate angle ambiguity. Section 4 analyzes the complexity of the proposed algorithm and the Cramer-Rao Bound (CRB). Section 5 is dedicated to conducting simulation analyses. This paper is concluded in Section 6.

Notations: Lowercase letters a, lowercase letters in boldface a, uppercase letters in boldface A, and letters in blackboard boldface A are used to represent scalars, vectors, matrices, and sets, respectively. ${\left[n^{\left(1\right)},n^{\left(2\right)},\cdots ,n^{\left(R\right)}\right]}^{T}$ represents an R-dimensional vector n, where $n^{\left(r\right)}$ is the rth coordinate. The transpose, complex conjugate, and complex conjugate transpose of A are $A^{T}$, $A^{*}$, and $A^{H}$. Tr[⋅] denotes the trace operator for a matrix. diag(⋅) represents the matrix formed by the diagonal elements of the matrix. ${\left[A\right]}_{i,j}$ denotes the (i,j) entry of A. 

## 2. Mathematical Model

The mathematical model of the coprime array is shown in Figure 1, which consists of two sparsely uniform linear arrays (SULAs) with M and  N sensors, respectively, where the inter-sensor spacings are Nd and Md (M and N are coprime, d is half a wavelength, M<N). The reference sensor is the leftmost sensor, and the reference sensors of both SULAs coincide. The location of the array can be denoted as [26]
(1)$$S_{CA}=\left\{Nxd|0\leq x\leq M-1\right\}\cup \left\{Myd|0\leq y\leq N-1\right\}.$$

In this paper, the enhanced coprime array (Figure 2) is obtained by increasing the number of sensors of each SULA within the coprime array. The number of sensors of the two SULAs is M+m and N+n, respectively. The other settings are consistent with the coprime linear array, and the location of the enhanced coprime array can be written as [27]
(2)$$S_{ECA}=\left\{Nxd|0\leq x\leq M+m-1\right\}\cup \left\{Myd|0\leq y\leq N+n-1\right\}$$

Denote $l_{i},i=1,2,\cdots ,G$ as the ith sensor in the enhanced coprime array and G=M+N+m+n−1.

Given an additive white Gaussian noise model with zero mean for the noise produced by the array upon receiving the signal, K far-field narrowband coherent signals, each coming from a different direction, are represented by $\theta ={\left[\theta_{1},\theta_{2},\cdots ,\theta_{K}\right]}^{T}$ and impinge on the aforementioned array. We can express the signal received from the array as [1]
(3)$$\begin{array}{c} x\left(t\right)=\sum_{k=1}^{K}\alpha_{k}a\left(\theta_{k}\right)s\left(t\right)+n\left(t\right)\\ =A\alpha s\left(t\right)+n\left(t\right), \end{array}$$
where $A=\left[a\left(\theta_{1}\right),a\left(\theta_{2}\right),\cdots ,a\left(\theta_{K}\right)\right]$ represents the direction matrix of the enhanced coprime array with $a\left(\theta_{i}\right)={\left[1,e^{j2\pi l_{2}\sin\theta_{i}/\lambda },\cdots ,e^{j2\pi l_{\left|S\right|}\sin\theta_{i}/\lambda }\right]}^{T}$. λ stands for the wavelength of the signal. s(t) is the waveform of the signal. The vector of the nonzero coherence coefficient is denoted by $\alpha ={\left[\alpha_{1},\alpha_{2},\cdots ,\alpha_{K}\right]}^{T}$. Additive white Gaussian noise is known as n(t). 

Supposing that the enhanced coprime array receives the signal of J snapshots, the model of the received signal can be written as [1]
(4)X=Aαs+N
where the signal waveform vector is denoted by s∈ℂ1×J. N represents the additive white Gaussian noise.

## 3. Proposed Algorithm

This section begins by presenting the spatial smoothing technique and the enhanced spatial smoothing technique. It then demonstrates the utilization of the prime characteristic to obtain the DOA without angle ambiguity. Finally, it explores the combination of the aforementioned methods to resolve the DOA of coherent signals received by the enhanced coprime array.

### 3.1. Spatial Smoothing Technique

The spatial smoothing technique [15] is capable of performing decorrelation processing by reducing the array aperture. By extending the coprime array, the impact of array aperture loss can be minimized. For the ith SULA of the extended mutual prime linear array, the received signal model can be expressed as
(5)$$X_{Si}=A_{Si}\alpha s+N_{Si}$$
where $A_{Si}$ and $N_{Si}$ correspond to the direction matrix of the ith SULA and the corresponding additive Gaussian white noise, respectively. Spatial smoothing techniques treat this SULA as consisting of multiple overlapping sub-arrays, as shown in Figure 3.

Denote the direction matrix of the kth subarray as $A_{k}$. According to the characteristics of the SULA, the relation between the direction matrix of the k+tth subarray and the kth subarray can be written as
(6)$$A_{k+t}=A_{k}D^{t}$$
where $D=diag\left(e^{j2\pi T\sin\theta_{1}/\lambda },e^{j2\pi T\sin\theta_{2}/\lambda },\cdots ,e^{j2\pi T\sin\theta_{K}/\lambda }\right)$, T is the sensor spacing of the ith SULA. Then, the received signal of the kth subarray can be modeled as
(7)$$\begin{array}{c} X_{k}^{}=A_{k}\alpha s+N_{k}\\ =A_{1}D^{k-1}\alpha s+N_{k}, \end{array}$$
where $A_{1}$ represents the direction matrix of subarray 1, and $N_{k}$ is the additive Gaussian white noise corresponding to the received signal of the kth subarray.

If the received signals are coherent, the rank of the covariance matrix of the received signal will be lower than the number of sources. Consequently, subspace algorithms such as MUSIC will fail to accurately estimate the DOA. To address this issue, we can apply the spatial smoothing technique to the covariance matrix, which restores the rank of the covariance matrix by averaging the covariance matrix of each subarray. The specific formulas of spatial smoothing technique can be modeled by
(8)$$\begin{array}{c} R_{SS}^{f}=\frac{1}{L}\sum_{k=1}^{L}R_{k}\\ \;\;\;\;=\frac{1}{L}\sum_{k=1}^{L}X_{k}^{}X_{k}^{H}/J\\ \;\;\;\;\;\;\;\;\;\;\;\;\;\;=\frac{1}{JL}\sum_{k=1}^{L}A_{1}D^{k-1}\alpha ss^{H}\alpha^{H}{\left(D^{k-1}\right)}^{H}A_{1}^{H}+\sigma_{}^{2}I\\ \;\;\;\;\;\;\;\;\;\;\;\;\;\;\;\;=A_{1}\left(\frac{1}{JL}\sum_{k=1}^{L}D^{k-1}\alpha ss^{H}\alpha^{H}{\left(D^{k-1}\right)}^{H}\right)A_{1}^{H}+\sigma_{}^{2}I\\ \;\;\;\;\;\;=A_{1}R_{S1}A_{1}^{H}+\sigma_{k}^{2}I, \end{array}$$
where $R_{S1}=\frac{1}{JL}\sum_{k=1}^{L}D^{i-1}\alpha ss^{H}\alpha^{H}{\left(D^{i-1}\right)}^{H}$. $\sigma^{2}$ means the average power of the Gaussian white noise. I represents the unit matrix.

According to [15], the number of sensors of the subarray needs to be larger than the number of coherent sources; that is, M>K. In the enhanced coprime array proposed in this paper, we can set m=n and choose the smoothing number as m. In this case, we can estimate, at most, M−1 sources. After that, we can estimate the DOAs in Equation (8) using the MUSIC algorithm.

### 3.2. Enhanced Spatial Smoothing Technique

The proposed approach in this section is enhanced spatial smoothing [24], which recovers the rank of the covariance matrix by making use of the signal subspace. Taking the example of the SULA with M+m sensors, the corresponding covariance matrix can be modeled as
(9)$$R=X_{M+m}X_{M+m}^{H}/J$$
where $X_{M+m}=A_{M+m}\alpha s+N_{M+m}$. $A_{M+m}^{}$ is the direction matrix of $X_{M+m}$, and $N_{M+m}$ denotes the corresponding white noise. 

By performing eigenvalue decomposition on Equation (9), we derive [20]
(10)$$\begin{array}{c} R\;=X_{M+m}X_{M+m}^{H}/J\\ =\Gamma_{s}+\Gamma_{n}\\ \;\;\;\;=\lambda_{s}u_{s}^{}u_{s}^{H}+U_{n}\Lambda_{n}U_{n}^{H} \end{array}$$
where $\lambda_{s}$ is the maximum eigenvalue of R, and $u_{s}$ denotes the corresponding eigenvector. $\Lambda_{n}$ represents a diagonal matrix including the rest of the M+m−1 smaller eigenvalues, and $U_{n}$ means the matrix consisting of the corresponding eigenvectors of these eigenvalues. 

For [4]: (11)$$u_{s}^{}u_{s}^{H}+U_{n}U_{n}^{H}=I_{M+m}$$
by applying Equation (11) to Equation (10), we derive
(12)AM+mRSAM+mH=R−σ2IM+m            =λsususH+σ2UnUnH−σ2IM+m       =(λs−σ2)ususH,
where IM+m∈ℂ(M+m)×(M+m) is the unit matrix and RS=αssHαH/J. 

Then, us can be written as
(13)    us=1λs−σ2AM+mSAM+mHus=AM+mt,
where t=1λs−σ2SAM+mHus∈ℂK×1. 

Denote M consecutive $u_{s}$ elements as
(14)$$\begin{array}{c} v_{i}=u_{s}\left(i:i+M-1\right)\\ \;\;\;\;\;=A_{M}D^{i-1}t,i=1,2,\cdots ,m+1. \end{array}$$
where $A_{M}$ means the direction matrix. Therefore, we can consider $v_{i}$ as the sub-array of spatial smoothing. Denote
(15)$$\begin{array}{c} \Gamma_{ij}=v_{i}v_{j}^{H}\\ \;\;\;\;\;\;\;\;\;\;\;\;=u_{s}\left(i:i+M-1\right)u_{s}^{H}\left(j:j+M-1\right)\\ \;\;\;\;\;\;\;\;=A_{M}D^{i-1}tt^{H}{\left(D^{j-1}\right)}^{H}A_{M}^{H}. \end{array}$$

Regard $\Gamma_{ij}$ as the covariance matrix, and the direction matrix is $A_{M}$. Performing a similar step as for spatial smoothing leads to
(16)$$R_{ESS}=\frac{1}{m}\sum_{i=1}^{m}\sum_{j=1}^{m}\Gamma_{ij}\Gamma_{ji}.$$

We can expand $R_{ESS}$ to obtain
(17)$$\begin{array}{c} R_{ESS}=\frac{1}{L}\sum_{i=1}^{L}\sum_{j=1}^{L}\Gamma_{ij}\Gamma_{ji}\\ \;\;\;\;\;=\frac{1}{L}\sum_{i=1}^{L}\sum_{j=1}^{L}\left(v_{i}v_{j}^{H}v_{j}v_{i}^{H}\right)\\ \;\;\;\;\;\;\;\;\;\;\;\;\;\;\;\;\;\;\;=\frac{1}{L}\sum_{i=1}^{L}\sum_{j=1}^{L}\left(A_{M}D^{i-1}tt^{H}{\left(D^{j-1}\right)}^{H}A_{M}^{H}A_{M}^{}D^{j-1}tt^{H}{\left(D^{i-1}\right)}^{H}A_{M}^{H}\right)\\ \;\;\;\;\;\;\;\;\;\;\;\;\;\;\;\;\;\;\;\;=A_{M}\left[\frac{1}{L}\sum_{i=1}^{L}\sum_{j=1}^{L}\left(D^{i-1}tt^{H}{\left(D^{j-1}\right)}^{H}A_{M}^{H}A_{M}D^{j-1}tt^{H}{\left(D^{i-1}\right)}^{H}\right)\right]A_{M}^{H}\\ \;\;=A_{M}R_{S2}A_{M}^{H}, \end{array}$$
where $R_{S2}=\frac{1}{L}\sum_{i=1}^{L}\sum_{j=1}^{L}\left(D^{i-1}tt^{H}{\left(D^{j-1}\right)}^{H}A_{M}^{H}A_{M}D^{j-1}tt^{H}{\left(D^{i-1}\right)}^{H}\right)$.

Then the DOA of the coherent signals can be estimated by applying the MUSIC algorithm to Equation (17).

### 3.3. Eliminating Angle Ambiguity Using Coprime Characteristics

DOA estimation using a SULA can result in angle ambiguity if the sensor spacing exceeds half a wavelength. In such scenarios, when only one source is present, multiple angle estimates for the arrival angles may be generated. To resolve this issue and obtain accurate angle estimates, it is crucial to perform angle ambiguity resolution. When angle ambiguity exists in the SULA, we can obtain [4]
(18)$$a\left(\theta_{i}\right)=a\left(\theta_{i}^{’}\right),\theta_{i}\neq \theta_{i}^{’}$$
where $a\left(\theta_{i}\right)$ represents the direction vector corresponding to the actual incident angle $\theta_{i}$, and $a\left(\theta_{i}^{’}\right)$ represents the direction vector corresponding to the ambiguous angle.

Assuming a SULA with a reference sensor located at the origin and its sensors arranged along the positive x-axis, the direction vector can be represented as follows [24]
(19)$$\exp\left(-j2\pi d\sin\theta_{i}/\lambda \right)=\exp\left(-j2\pi d\sin\theta_{i}^{’}/\lambda \right)$$

Then we obtain
(20)$$2\pi d\sin\theta_{i}^{’}/\lambda -2\pi d\sin\theta_{i}/\lambda =2\alpha \pi$$
where α∈Z. Denote $d=m\lambda /2,m\in Z^{+}$, and the above equation can be written as
(21)$$\sin\theta_{i}^{’}-\sin\theta_{i}=\frac{2\alpha }{m}$$

Since the range of incident signal angles is $\left(-90^{\circ },90^{\circ }\right)$, then we can obtain $\sin\theta_{i}^{’}\in \left(-1,1\right)$, which means
(22)$$\frac{2\alpha }{m}+\sin\theta_{i}\in \left(-1,1\right)$$

By solving the above equation, we obtain
(23)$$\alpha \in \left(\frac{\left(-1-\sin\theta \right)m}{2},\frac{\left(1-\sin\theta \right)m}{2}\right)$$When there is no angle ambiguity (α=0), it corresponds to m≤1. When m>1, as indicated by the equation above, α has m different values. This implies that within the range of $\left(-90^{\circ },90^{\circ }\right)$, there exist m ambiguous angles, including the angle of the original signal of interest. Therefore, when using techniques like MUSIC and peak searching in the spectrum, you will obtain m peaks, and it is necessary to discard m−1 ambiguous angles.

Assuming that the receiving array is a non-uniform linear array with M sensors positioned along the x-axis, and $x=\lambda /2\cdot \left[0,x_{1},x_{2},\cdots ,x_{M-1}\right]$ represents the positions of the elements on the x-axis, where $x_{n}$ is a non-negative integer, then its direction vector can be represented as
(24)$$a\left(\theta_{i}\right)={\left[1,\exp\left(-j\pi x_{1}\sin\theta_{i}\right),\cdots ,\exp\left(-j\pi x_{M-1}\sin\theta_{i}\right)\right]}^{T}$$

The angle ambiguity is
(25)$$\sin\left(\theta_{i}\right)-\sin\left(\theta_{i}^{’}\right)=\frac{2\alpha_{n}}{x_{n}}$$
where $\alpha_{n}\in Z,x_{n}\in N^{+}$. 

If the array does not have angle ambiguity, we must have
(26)$$\frac{2\alpha_{n}}{x_{0}}\cap \frac{2\alpha_{n}}{x_{1}}\cap \cdots \cap \frac{2\alpha_{n}}{x_{M-1}}=0,\alpha_{n}\in Z.$$

Denote $\rho_{n}=\frac{2\alpha_{n}}{x_{n}}$, then over all possible values of $x_{n}$, we can obtain
(27)$$\rho_{n}=\left\{0,\frac{2\alpha_{n}}{x_{n}}\right\},|\alpha_{n}|<x_{n}.$$

When $x_{n}$ is prime, its greatest common divisor can only be 1, which means
(28)$$\left\{\rho_{0}\cap \rho_{1}\cap \cdots \cap \rho_{M-1}\right\}=\left\{0\right\}.$$

Equation (28) demonstrates that if the array positions $x_{n}$ are mutually prime, we can obtain $\sin\left(\theta_{i}\right)-\sin\left(\theta_{i}^{’}\right)=0$, i.e., $\theta_{i}=\theta_{i}^{’}$. This indicates that there is no issue of angle ambiguity in the estimation of DOA. Consequently, by ensuring that the sensor spacing satisfies the coprime relationships among them, it becomes possible to effectively overcome the problem of angle ambiguity that arises when the sensor spacing exceeds a half-wavelength.

The enhanced coprime array enhanced spatial smoothing MUSIC algorithm (ENSS-MUSIC), as demonstrated in Algorithm 1 of this paper, can be summarized as follows.
**Algorithm 1:** Enhanced spatial smoothing MUSIC algorithmStep 1: Calculate the covariance matrix of each SULA according to Equation (4);Step 2: Perform the eigenvalue decomposition of the covariance matrix according to Equation (10); $\mathrm{Step}\;3:\;\mathrm{Choose}\;\mathrm{the}\;\mathrm{eigenvector}\;\mathrm{that}\;\mathrm{corresponds}\;\mathrm{to}\;\mathrm{the}\;\mathrm{largest}\;\mathrm{eigenvalue}\;\mathrm{as}\;u_{s}$;Step 4: Calculate the new covariance matrix according to Equations (15) and (16); Step 5: Use the MUSIC algorithm to estimate the DOAs;Step 6: Compare the spectral peaks of subarray1 and subarray2 to find the angle corresponding to the common spectral peak, which is the true angle.

## 4. Performance Analysis

### 4.1. Computation Complexity Analysis

In this section, we undertake a comparison of the complexity of the proposed algorithm, along with several compressive sensing algorithms. This comparison includes the evaluation of algorithms, such as SBL and NNM. The complexity of SS-MUSIC is divided into two parts: SULA 1 and SULA 2. For SULA 1, the covariance matrix is calculated m times, and the number of multiplications is $M^{2}Jm$. The total number of multiplications using the MUSIC algorithm is $M^{3}+M^{2}\left(K+2\right)g$, and g denotes the number of searches. Similarly, for SULA 2, the total number of multiplications is $N^{2}Jn+N^{3}+N^{2}\left(K+2\right)g$. Therefore, the computation complexity of SS-MUSIC is
$$O\left(M^{3}+N^{3}+\left(M^{2}m+N^{2}n\right)J+\left(M^{2}+N^{2}\right)\left(K+2\right)g\right).$$

The computation complexity of ENSS-MUSIC is similar to that of SS-MUSIC, which is
$$O\left(M^{3}+N^{3}+{\left(M+m\right)}^{2}J+{\left(N+n\right)}^{2}J+{\left(M+m\right)}^{3}+{\left(N+n\right)}^{3}+M^{2}m+N^{2}n+\left(M^{2}+N^{2}\right)\left(K+2\right)g\right).$$

The selected algorithms for comparison are compressive sensing algorithms, known for their iterative convergence. Hence, their complexity is dependent on the number of iterations. Determining the appropriate number of iterations based on the desired error introduces complexity in calculating the overall complexity. To assess this, we compared the runtime for various numbers of iterations, as presented in the table. We conducted analyses using 1, 20, 100, and 1000 iterations to evaluate the time required for different iteration counts. See Table 1.

### 4.2. Cramer-Rao Bound (CRB)

According to [34], the Cramer-Rao Bound (CRB) matrix can be modeled as
(29)CRB=σn22J{Re[DHΠA⊥DP^T]}−1
where A represents the manifold matrix of the array, $\Pi_{A}^{⊥}=I-A{\left(A^{H}A\right)}^{-1}A^{H}$, $\hat{P}=1/J\sum_{t=1}^{J}s\left(t\right)s^{H}\left(t\right)$, $\sigma_{n}^{2}$ denotes the average power of signal source, and D can be written as
$$D=\left[\frac{\partial a\left(\theta_{1}\right)}{\partial \theta_{1}},\frac{\partial a\left(\theta_{2}\right)}{\partial \theta_{2}},\cdots ,\frac{\partial a\left(\theta_{K}\right)}{\partial \theta_{K}}\right]$$
where $a\left(\theta_{K}\right)$ denotes steering vector.

### 4.3. The Advantages of the Proposed Array and Algorithm

We propose an enhanced coprime array for the DOA estimation of coherent signals. Our design offers several advantages: 

By employing enhanced spatial smoothing techniques, the DOA can be estimated using subspace-based algorithms like MUSIC accurately. The enhanced coprime array can estimate M−1 coherent signals at most;The enhanced spatial smoothing technique effectively utilizes all the signal subspace, leading to improved accuracy in DOA estimation and enhanced resistance to noise interference;The designed enhanced spatial smoothing MUSIC algorithm offers lower computational complexity compared with the compressed sensing algorithm. Additionally, it provides higher DOA estimation accuracy, improving its suitability for real-world applications.

## 5. Simulation Results

In this study, we have carefully chosen several widely recognized compressive sensing algorithms, including SBL and NNM. These algorithms were thoroughly evaluated and compared to our own proposed algorithm, with a detailed analysis of their respective performances. For this evaluation, the parameters of the proposed array are fixed as M=3, N=4, x=y=1. In this setup, two far-field narrowband coherent signals impinge on the proposed array from angles of 9.55° and 25.15°. To quantify the accuracy of the direction of arrival (DOA) estimates, we defined the root mean square error (RMSE) as
(30)$$RMSE=\sqrt{\frac{1}{K}\frac{1}{J}\sum_{k=1}^{K}\sum_{j=1}^{J}{\left({\hat{\theta }}_{j,k}-\theta_{k}\right)}^{2}}$$
where J and K stand for the total number of coherent sources and Monte Carlo trials, respectively. The estimate of the actual angle $\theta_{k}$ is denoted by ${\hat{\theta }}_{j,k}$. We have J=1000 as the number of Monte Carlo simulations in the following simulations.

In Figure 4, we present a comparison of the spectral peak outcomes obtained from the MUSIC algorithm after applying enhanced spatial smoothing to the two subarrays. The SNR is set as 10 dB, and the number of snapshots is set as 500. Evidently, only the peaks aligning with the true DOA of the coherent signals are observed, consistent with our previous analysis. Figure 5 illustrates a scatter plot of the DOA estimation results across 100 iterations. The SNR is set as 0 dB and the number of snapshots is set as 500. It is apparent that the estimation results consistently form a straight line with minimal variability, suggesting that the proposed algorithm yields highly stable DOA estimates for coherent sources.

A comparison of the RMSE curves for the various algorithms, with differing SNRs, is shown in Figure 6. The simulation was conducted using 200 snapshots. Notably, the proposed algorithm outperforms other algorithms even under low SNR conditions. This superiority can be attributed to the effective utilization of an enhanced spatial smoothing technique, which incorporates information from the signal subspace, resulting in improved performance. As a result, the performance of the enhanced spatial smoothing technique surpasses that of the spatial smoothing technique. In Figure 7, we evaluate the RMSE of different algorithms with a varying number of snapshots while maintaining an SNR of 5 dB. It is evident that the RMSE of the proposed algorithm is smaller compared to that of other algorithms, indicating a lower estimation error and superior performance.

The curves of RMSE are compared in Figure 8 as the SNR varies for different numbers of array sensors. The simulation settings remain unchanged, except for the variation in the number of sensors. It is evident that as the SNR increases, the RMSE consistently decreases, indicating enhanced DOA estimation performance. Furthermore, increasing the number of array sensors also leads to a decrease in RMSE, suggesting that the performance of the proposed algorithm is enhanced by increasing the number of sensors.

The curves of RMSE are compared in Figure 9, as the SNR varies for different numbers of sources. The array parameter is fixed at M=5, N=6, m=3, n=3. It is evident that as the number of sources increases, there is a consistent rise in RMSE, implying a decrease in DOA estimation performance. Conversely, an improvement in SNR results in a decrease in RMSE, indicating an enhanced DOA estimation performance.

## 6. Conclusions

This paper introduces a novel enhanced coprime array designed for the DOA estimation of coherent signals. Additionally, it presents the DOA estimation algorithm tailored specifically for this array. The algorithm incorporates advanced spatial smoothing techniques to effectively handle coherent signals and exploit the signal subspace of received signals, resulting in enhanced DOA estimation performance and noise robustness. The complexity analysis of the proposed algorithm has been demonstrated to be lower than that of the compressed sensing algorithm. Numerous simulation experiments have been conducted to confirm that the proposed array effectively achieves the estimation of DOA for coherent sources. Furthermore, the performance of the proposed algorithm surpasses that of both the spatial smoothing MUSIC algorithm and the compressed sensing algorithm.

## Figures and Tables

**Figure 1 sensors-24-00260-f001:**
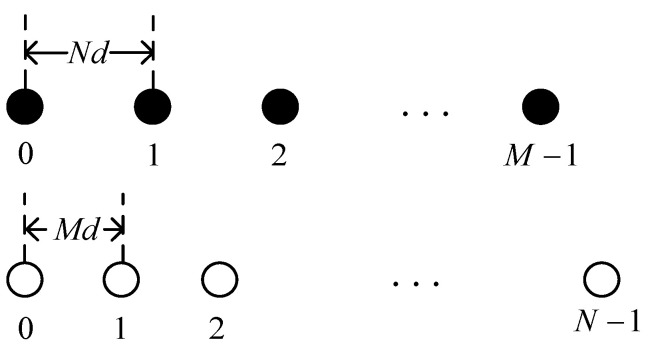
Coprime array.

**Figure 2 sensors-24-00260-f002:**
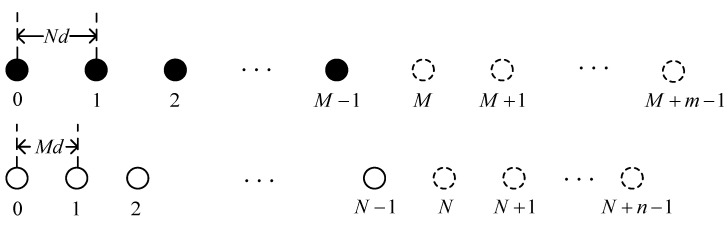
Enhanced coprime array.

**Figure 3 sensors-24-00260-f003:**
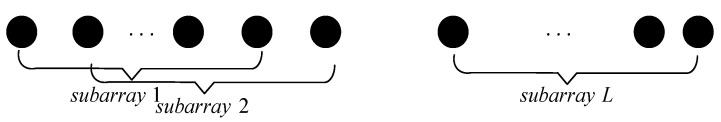
Spatial smoothing technique.

**Figure 4 sensors-24-00260-f004:**
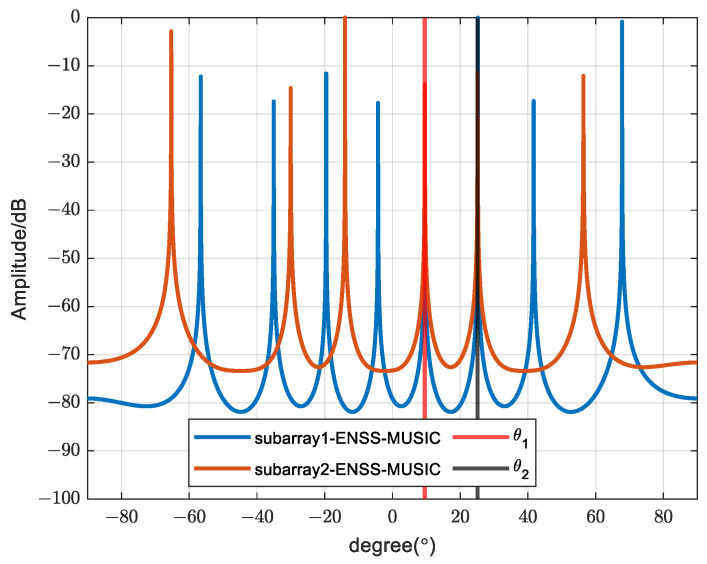
Spectrum peak comparison of different subarrays.

**Figure 5 sensors-24-00260-f005:**
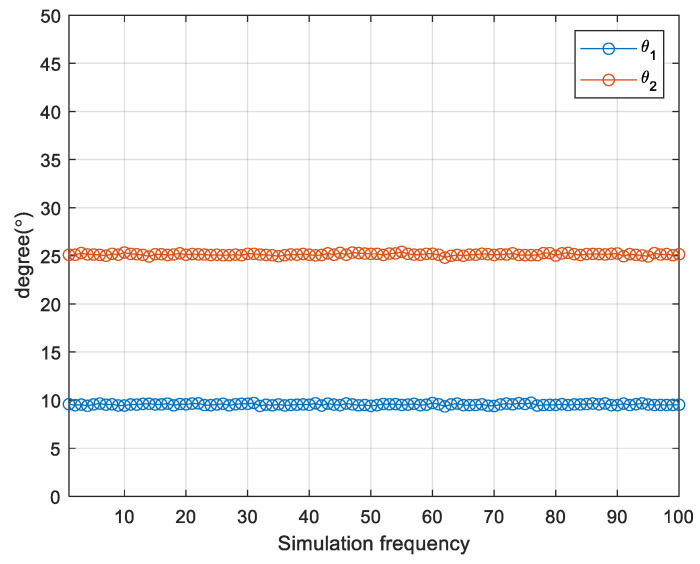
Scatter plot of DOA estimation results for 100 iterations.

**Figure 6 sensors-24-00260-f006:**
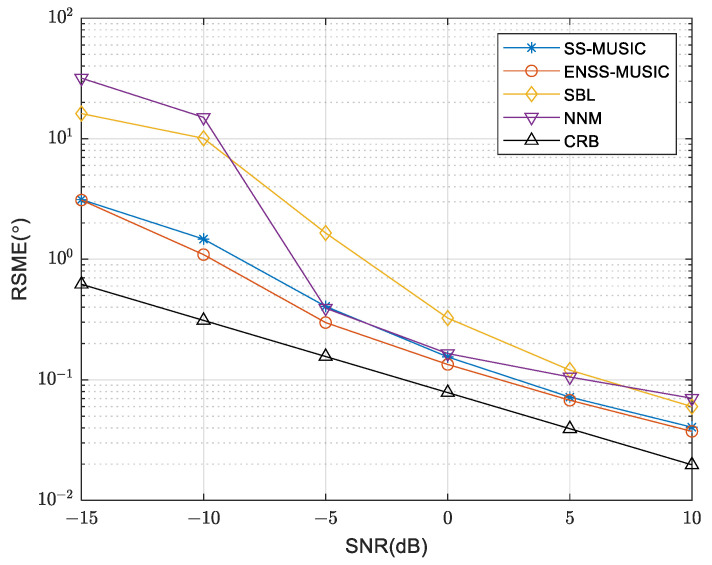
RMSE comparison of various methods using different SNRs.

**Figure 7 sensors-24-00260-f007:**
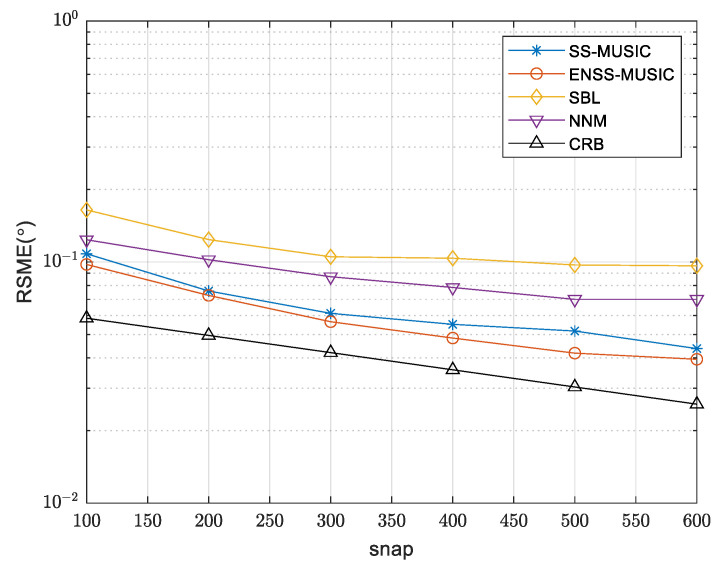
RMSE comparison of various methods with different numbers of snapshots.

**Figure 8 sensors-24-00260-f008:**
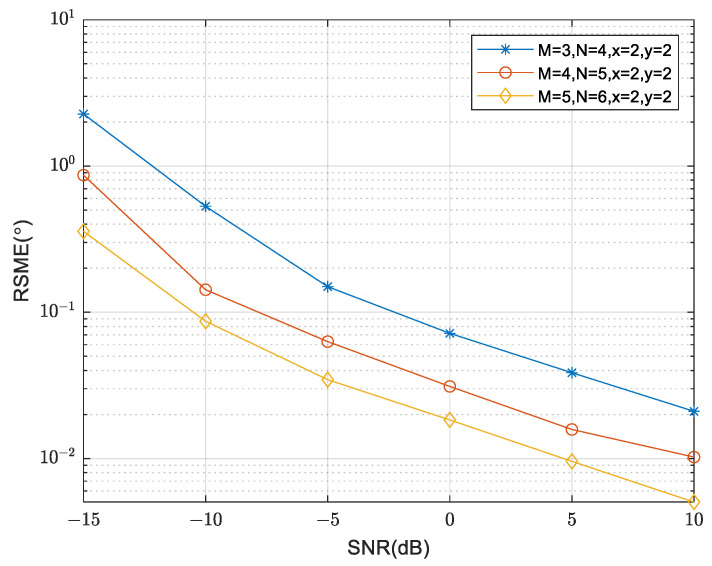
Comparison of RMSE under different numbers of array sensors.

**Figure 9 sensors-24-00260-f009:**
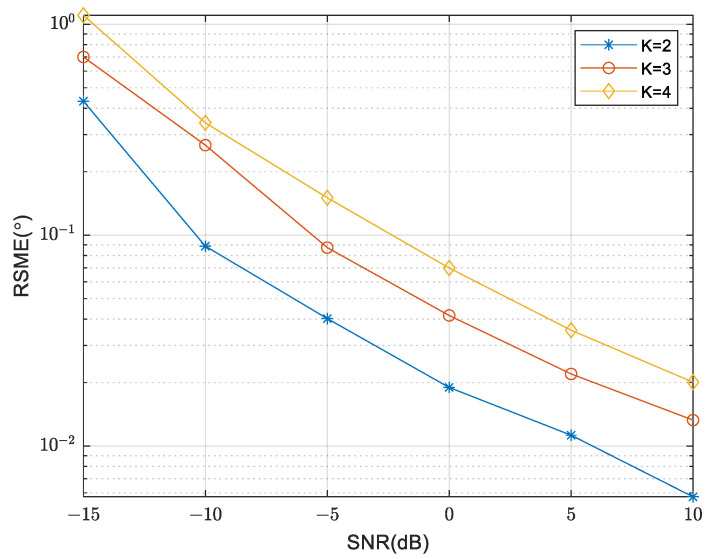
Comparison of RMSE under different numbers of sources.

**Table 1 sensors-24-00260-t001:** Comparison of runtime of different algorithms.

Monte Carlo	NNM	SBL	SS-MUSIC	ENSS-MUSIC
1	1.3343 s	0.2013 s	0.0572 s	0.1311 s
20	5.2786 s	3.7636 s	0.7131 s	0.7262 s
100	23.5879 s	17.4307 s	3.7138 s	3.5957 s
1000	254.0145 s	175.8712 s	36.3054 s	36.4978 s

## Data Availability

Data are contained within the article.

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
