# Peer review of "Enhanced Coprime Array Structure and DOA Estimation Algorithm for Coherent Sources"

_sensors, 2024, doi:10.3390/s24010260_

Round 1

Reviewer 1 Report

Comments and Suggestions for Authors

This paper considers the problem of DOA estimation of coherent sources using coprime array. While the presented technical content seem solid, I believe that the manuscript should be enhanced according to the following comments before acceptance.

- The descriptions in the Abstract and the Contribution Summary of Introduction should be more specific --- how the coprime array and spatial smoothing algorithm are enhanced should be briefly explained.

- The rationale behind the idea, or, why the proposed method outperforms the existing approaches should be justified.

- The old-dated references published more than five years ago should be replaced by discussion of more recent and high relevant references such as [r1] Rank minimization-based Toeplitz reconstruction for DoA estimation using coprime array, IEEE Communications Letters, vol. 25, no. 7, pp. 2265–2269, July 2021.

- In the legend of Figure 7, the CRB is missing.

Comments on the Quality of English Language

Minor editing of English language required.

Author Response

Dear Editor and Reviewers,

  Thank you very much for your Email, and many thanks for coordination with our paper.

First of all, the authors would like to thank the associate editor and the reviewers for the time and effort they have put into our paper. We have carefully read all of your comments and have made necessary modifications to our revised manuscript. The current manuscript has been revised and refreshed by our co-authors. According to the reviewers’ comments, we hereby write a revision report and list the following modifications.

 We are uploading our point-by-point response to the comments (Revision Report).

Reviewer 2 Report

Comments and Suggestions for Authors

The authors present an enhanced coprime array for the direction of arrival (DOA) estimation. The work is of great interest. The following comments should be addressed before further processing.

1) The introduction section can be improved by adding RECENT literature, such as

a) “Deep Learning Approach to Source Localization of Electromagnetic Waves in the Presence of Various Sources and Noise,” Symmetry, vol.15, 1534, pp. 1-15, 2023.

b) “Spherical atomic norm-inspired approach for direction-of-arrival estimation of EM waves impinging on spherical antenna array with undefined mutual coupling,” Applied Sciences, vol.13, no.5, pp. 1-16, 2023.

etc.

2) Equations that are not from the authors should be properly cited. For instance, Equation 2.

3) Figures and values of performance should be added to the Abstract and Conclusion sections.

4) The paper needs english proofreading.

Comments on the Quality of English Language

Nil

Author Response

(The authors gave the same response as above.)

Reviewer 3 Report

Comments and Suggestions for Authors

This work proposed a new enhanced coprime array for the direction of arrival (DOA) estimation. Through the proposed enhanced coprime array, spatial smoothing techniques can be applied to process coherent sources by increasing the number of array sensors within the coprime array. This reviewer thinks that this work needs to be revised based on following comments.

1.       The order of matrix multiplication is very important. It is needed to check the third line of equation (17), because it is different from the matrix product shown in equation (15). If it doesn't matter if the matrix order is changed, additional explanation should be needed to understand.

2.       Since ‘wavenumber (2p/l)’ is generally expressed as ‘k’, it is better to replace ‘k’ notation with another notation in equation (20).

3.       It is necessary to check for missing expressions or typo. For example, the parentheses are missing in ‘–90°, 90°’ below equation (21), and there is a typo in the Introduction: ‘ROOT-MUISC’.

4.       It is needed to check the RMSE equation (30). There is no ‘i’ inside the sigma function. Also, the parentheses are omitted.

5.       In Figure 7, the legend for the black line is missing. It should be added.

6.       It is needed to check the format of the figure caption. The caption starts with an uppercase.

7.       In order to make it easier to understand the proposed optimal coprime array structure, it would be better to add an array structure like the array shape presented in the following paper:

[1] S. Qin, Y. D. Zhang and M. G. Amin, “Generalized Coprime Array Configurations for Direction-of-Arrival Estimation,” in IEEE Transactions on Signal Processing, vol. 63, no. 6, pp. 1377-1390, March 15, 2015, doi: 10.1109/TSP.2015.2393838.”

[2] W. Zheng, X. Zhang, Y. Wang, M. Zhou and Q. Wu, “Extended Coprime Array Configuration Generating Large-Scale Antenna Co-Array in Massive MIMO System,” in IEEE Transactions on Vehicular Technology, vol. 68, no. 8, pp. 7841-7853, Aug. 2019, doi: 10.1109/TVT.2019.2925528.”

8.       Figure 5 shows the DOA estimation results for 100 times using the proposed method. This is a good result because it is very similar to the target value, but it looks like the same values were plotted. I think it would be better to adjust the scale to show that there are some differences.

Author Response

(The authors gave the same response as above.)

Round 2

Reviewer 2 Report

Comments and Suggestions for Authors

My comments have been addressed.

Comments on the Quality of English Language

Nil